# Wear Mechanisms and Wear Model of Carbide Tools during Dry Drilling of CFRP/TiAl6V4 Stacks

**DOI:** 10.3390/ma12182843

**Published:** 2019-09-04

**Authors:** Unai Alonso Pinillos, Severo Raúl Fernández Vidal, Madalina Calamaz, Franck Andrés Girot Mata

**Affiliations:** 1Department of Mechanical Engineering, University of the Basque Country (UPV/EHU), 48013 Bilbao, Spain; 2Department of Mechanical Engineering & Industrial Design, Faculty of Engineering, University of Cadiz, Av. Universidad de Cadiz 10, E-11519 Puerto Real-Cadiz, Spain; 3Arts et Métiers, I2M, UMR 5295, Esplanade des Arts et Métiers, F-33405 Talence CEDEX, France; 4IKEBASQUE, Basque Foundation for Science, E-48013 Bilbao, Spain

**Keywords:** carbide wear, CFRP, titanium alloy, abrasion, metal adhesion

## Abstract

The present contribution on tool wear during the drilling of carbon fiber composite materials (CFRP)/Ti stacks intends to determine (i) if the adhesion of titanium to carbide is mechanical or chemical, (ii) the possible diffusion path, (iii) if the titanium is the only element involved in the adhesion and (iv) the role of the CFRP in this wear. The overall tool wear is not the sum of the wear in each material and there is a multiplicative effect between them. It has been pointed out that the maximum temperature reached during drilling is higher than 180 °C, 400 °C and 750 °C respectively in the CFRP and Ti plates alone and in the Ti part of the stack. As tungsten carbide CW is not in equilibrium with titanium above 250 °C, the diffusion path is CW/(Ti,W)C/Ti as confirmed by Auger analysis. For temperatures above 500 °C, (Ti,W)C becomes very sensitive to oxidation allowing a friable oxycarbide (Ti,C,O) to form, which explains the erosion of the tool. The CW is therefore the weakest link in the drilling of CFRP/Ti stacks. Improving the performance of the tool involves the use of a coating, the development of a tool material having low chemical affinity with Ti and/or the use of cryogenic lubricant.

## 1. Introduction

### 1.1. Importance of Tool Wear in Aerostructure Drilling: Material Damages and Tooling Costs

In the last years, aeronautical manufacturing, and in particular structural assembly, has undergone profound transformations towards ever higher levels of automation. The crossroad is obvious: automate processes or relocate production. The reasons for thinking about automation are several: cost reduction, quality improvement, and increased safety at work.

The weight of manual activity in the aeronautical assembly remains high and represents between 25% and 75% of the total cost of the structure. All possibilities for improvement in these operations translate into savings opportunities and among them the automation of elementary processes (drilling, sealing, riveting...) or even the entire process are options that are beginning to be consolidated.

Traditionally, the operations of joining elements such as skins and fairings to the internal structure formed by spars and ribs have been performed manually or semi-automatically, running thousands of holes in each element. In these types of joints, very precise tolerances of the position and geometry of the holes are required. 

With the development and use of carbon fiber composite materials (CFRP), drilling systems are needed that can be adapted to the different materials that constitute the metal/composite structure (CFRP, titanium alloy Ti6% Al4% V, alloys of aluminum 2XXX or 7XXX). Currently, with semi-automatic systems, the entire stack is drilled at the most restrictive speed and feed rate—which is the rate of titanium—but these drilling conditions may not be suitable for the CFRP because it can leave a bad finish of the hole and quickly wear the tool.

The understanding of the wear mechanisms and the behavior of the carbide drill in the metal/composite stacks is therefore essential, because it entails a cost in tools that can quickly be very expensive.

### 1.2. Wear Mechanisms in CFRP, Ti Alloys and CFRP/Ti Alloys Stacks

Xu et al. [1] defined the wear types of carbide tools during CFRP/Ti stack drilling. They consist of an interaction between abrasive wear and adhesion wear. These mechanisms generate, at the level of the active parts the tool, wear of the edges and flanks, a local withdrawal of the cutting edge and the phenomena of the micro-chipping and micro-fracture of the cutting edges or of the lands of the drill.

In CFRP, it is now well understood that abrasion is the main cause of wear for carbide tools, leading to edge and flank wear and a rounding of the cutting edge [2,3,4,5,6,7,8,9].

For adhesion wear in the Ti-6%Al-4%V alloy, Calamaz, Girot, Ezugwu et al. [10,11,12] proposed that titanium can react with:(i)Tungsten carbide CW to form a solid solution (Ti,W)C, or TiC carbide type, as confirmed by the pseudo-binary CW-Ti diagram (Figure 1a),(ii)cobalt to allow the formation of Ti_2_Co, TiCo, TiCo_2_ or TiCo_3_ compounds [13](iii)or also to form a solid solution of Ti in α-Co (Figure 1b).

This interaction between tool and chip is strongly influenced by the chip morphology (continuous or serrated) as pointed out by Calamaz et al. [10] and therefore by the thermal conditions at the chip/tool interface.

In a study on friction stir welding—a process comparable with drilling in terms of loads, shearing rates and temperature—Wang et al [14] demonstrated the existence of a reaction zone between the work piece and tool material, with a thickness in the range 100–400 nm. The reaction layer was composed of nano-sized grains, extending into the Ti-6Al-4V alloy and adhering to the tool.

Li et al. [15] showed that new W2C, W and TiC phases could be formed in this zone. They proposed the following reactions between titanium and tungsten carbide:
WC + Ti →TiC + W   and/or    2WC + Ti → W_2_C + TiC

Thermodynamic calculations showed a decrease in free energy for these reactions when the temperature varied from 482 °C (900 K) to 982 °C (1800 K), suggesting that W and W_2_C products are thermodynamically possible.

Various authors have attempted to establish the wear mechanisms of tungsten carbide tools during the drilling of CFRP/Ti-6% Al-4V stacks [16,17,18,19,20,21,22]. According to Park et al. [16], Ti adhesion is a predominant wear factor in the drill cutting edges. The higher the cutting speed (and thus the speed rotation of the drill), the higher the tool wear, due to the higher temperature generated, especially during the drilling phases of the Ti plates. Park et al. [17] also found that the CWtool was subject to significant abrasion and edge damage during CFRP drilling, as well as significant sidewall wear, adhesion and abrasion wear when drilling the Ti. On the other hand, the physicochemical mechanisms at the origin of these wears have not yet been explained.

In the same way, Poutord et al. [18] investigated the local wear of uncoated K20 type carbide drills and confirmed the high Ti adhesion on the drill faces and rounded wear of the main cutting edges.

Concerning CFRP/aluminum stacks, Zitoune et al. [23] investigated the impact of the machining parameters on cutting forces, holes’ quality and the CFRP/Al interface for the twist drill and double cone drill. They pointed out that the double cone drills induce less thrust force compared to the standard twist drill. A numerical analysis of the drilling process enabled the critical thrust force responsible for the delamination of the last ply to be related to the maximum thrust force responsible for the interface separation of CFRP/Al from aluminum plate thickness.

Sorrentino et al. [24] analyzed the problem of damage due to the high temperatures reached during the CFRP dry drilling process, using K type thermocouples positioned in the workpiece, near the hole surface. They pointed out that the maximum temperature reached near the hole surface decreased with the increase of feed speed or the decrease of cutting speed. Temperatures between 70 and 90 °C have been measured (at 1 mm from the hole surface).

These different studies demonstrate that although a lot of work has been done on the subject, the origin of the wear mechanisms in the metal/composite stacks are still poorly known. We do not know (i) if the adhesion of titanium to carbide is only mechanical or chemical; (ii) in this last case, if the possible diffusion path is through the carbide or the cobalt binder; (iii) if the titanium is the only element involved; (iv) what role the CFRP has in this wear, (v) if the overall wear of the tool is the sum of the wear in each of the materials or if they have a multiplicative effect between them.

This knowledge of the modes of wear is important to improve the tool materials for this type of application and thus to avoid the use of coatings which are often expensive.

### 1.3. Wear Models for Prediction of Tool Life

Wear is generally defined as any change in the shape of the active part of a tool, relative to its initial shape, resulting from the progressive loss of tool material during cutting. Different wear types have been defined by standard. In order to determine the tool lifetime and to compare the influence of different cutting parameters, it is necessary to use as a criterion a defined type of damage of the active part. The life criterion can be a predetermined numerical value of any type of tool damage that can be measured. When more than one damage form becomes measurable, each of them should be evaluated and the end of life of the tool is reached when any of the damage limits are achieved. It is also possible to define more simply the damage of a cutting tool in an indirect way by using as a wear criterion the surface roughness or the geometrical tolerances of the parts. In the drilling of CFRP, for example, the drill life can be defined by the number of holes drilled in a required quality or the occurrence of delamination. From the wear criteria, it is possible to establish life models. Modified Taylor or Taylor models are thus among the oldest and most used. They connect the tool service life to the cutting parameters (cutting speed, feed rate, depth of cut, etc.). In the literature, however, very few authors have introduced wear into their model.

The model of Tsao et al. [25] takes into account the wear of the tool (excluding the essential machining parameters such as feed rate, cutting speed, etc.). It links the feed force with tool wear and the damage of the material, using as a criterion the maximum allowable force to ensure a certain quality of machining. Lin and Ting [26] proposed a wear model based on the control of the feed force F and the milling or drilling torque M as a function of the cutting parameters (cutting speed Vc (m/min), feed rate f (mm/rev) or feed rate Vf (mm/min), tool diameter d and wear W). The authors also highlighted a greater sensitivity of the feed force than the torque to the variations of the wear of the tool.

Among the phenomenological models, that of Iliescu et al. [2] seems the most successful. It is based on the evolution of the feed force with tool wear *W* and the cutting conditions: feed rate *f*, cutting speed *Vc*. Terms can be added to take into account the geometry of the tool [27]. The tool wear is based on the Archard model, which takes into account the force applied and the relative velocity between the wearer and the worn body. The speed can be defined as the length *Lc* traveled by the tool tip in contact with the composite material divided by the machining time. The wear is therefore proportional to the product of the feed force by the contact length, different from the machined length. During a test *i*, all the machining mechanisms related to the machining sequences between *i* = 1 and *i* − 1 must therefore be taken into account. The axial force *F_ai_* for the *i^th^* machining sequence will then depend on the value of the tool wear of the consecutive *i* − 1 previous machining steps.
(1)$$F_{ai}=K\cdot f_{i}^{\alpha }\;\cdot \;V_{C\;i}^{\beta }\;\cdot \;g(W_{i-1})$$
where *g*(*W_i_*) is a function of the tool wear and is defined as follows for uncoated carbide tools:
(2)$$g(W_{i})={\left(W_{0}+A_{0}\;\cdot \;W_{i-1}\right)}^{\delta }$$
(3)$$W_{i-1}=\sum_{j=1}^{i-1}F_{aj}\;\cdot \;L_{Cj}$$
with
(4)$$L_{C}=\pi \;\cdot \;d\;\cdot \;\frac{e}{f}$$

In that model, *W*_0_ represents the initial value of cutting edge sharpness, *A*_0_ the tool abrasion rate, *g*(*W_i_*) the tool wear after the *i^th^* test and *Kc* a constant depending on the geometry of the tool and the properties of the material being machined.

Girot et al. [28] have also adapted this model to the drilling of metals, in dry drilling conditions. The phenomenon of built-up layer (BUL) adhesion is the predominant tool damage mechanism.

For this aspect, there is also a lack of models to predict when the drill has suffered too much wear to affect the quality of the machined surface.

### 1.4. Work Developed

The present contribution aims at elucidating (i) the origin of the wear mechanisms in the metal/ composite stacks during drilling, (ii) the possible diffusion path through the carbide or the cobalt binder if diffusion is involved, (iii) the role of each material (CFRP and Ti alloy) in the overall wear and at defining (iv) wear models which enable predictions of this wear for process control and monitoring.

## 2. Methodology

### 2.1. Drill Selection

The tool selected is a conventional micro grain carbide SECO twist drill with 6%Co as a binder, adapted for the drilling of non-ferrous materials. The dimensions and geometry of the drill are showed in Figure 2.

### 2.2. CFRP, Ti and CFRP/Ti6Al4V Materials

The specimens tested in this study are T300 carbon fiber composites of 6 and 10 mm thick and Ti-6%Al-4%V plates of 5 mm. The CFRP material was manufactured by Titania (Cadiz, Spain) using T300 carbon fiber/AIMS05-27-002 epoxy prepregs and autoclave polymerization. Both have been tested individually to understand the mechanisms of wear in each material, as well as drilled jointly and rigidly tied together, to understand the phenomenon of wear in the stacks (Figure 3). The CFRP/titanium stack is neither bonded nor co-cured, but rigidly fixed by screws during its positioning on the machining center.

### 2.3. Drilling Procedure and Equipment Used

The drilling sequence has been CFRP/Ti6Al4V, as is generally done in the industrial field. This is due to the tendency of the composite material to undergo delamination at the exit. Because this phenomenon occurs by the rupture of the last layer of CFRP—a consequence of the axial stress generated by the tool—the placement of the titanium sheet below the composite eliminates that phenomenon. Due to the need to obtain a noticeable wear in a reasonable number of holes, aggressive cutting parameters were chosen for the tool, in terms of feed rate and cutting speed. For this reason, the cutting conditions used to analyze wear have been chosen in external sources, like other research works or real application cases, and defined in Table 1. The machine used for the tests is a three-axis Kondia A-6 machining center with a FAGOR 8070 numerical control.

Two types of tests were carried out. Wear tests on the 3 types of materials (CFRP, Ti and stack) and tests interrupted after 10, 25, 40 and 60 holes on the stack. The purpose of these tests was to determine the amount of wear associated with drilling in the metal and in the CFRP.

### 2.4. Data Recorded (Axial Load, Torque, Temperature, Wear)

During the drilling tests, loads, torque and temperature have been recorded. Loads and torque have been measured using a 3 components KISTLER© 9257B dynamometer (Fx, Fy and Fz) and a 2 components Artis dynamometer (Fz and Mz). These dynamometers are connected to a transducer and signal amplifier, which transfers the acquired data to the computer using LabView software for later processing. The sampling frequency set in the software is 1000 Hz.

In order to know the temperature range supported by the drill in each material during drilling, temperature measurements have been carried out in different areas of the workpiece and the tool. Holes have been made leaving a thickness of 1 mm between the face where the temperature is to be measured and the hole (Figure 4). With this configuration, the temperatures in the composite material and the titanium during drilling have been measured using an OPTRIS PI160 thermographic camera and an IMPAC IGAR 12-LO pyrometer [29]. Prior to carrying out the measurements, the emissivity of each of the materials has been calibrated by comparison with the measurements of the pyrometer until obtaining temperature signals at the measurement point with an error less than 1% between the pyrometer and the thermographic camera. The determined emissivity was 0.9 and 0.35 in the CFRP and the titanium alloy, respectively.

Drill wear is measured using a KEYENCE VHX-1000E digital microscope which allows a 2D profile of the cutting edge to be obtained by the changes of texture of the picture recorded (Depth from Defocus). These measurements are performed perpendicularly to the two main edges and at three points: 200 μm from the tip of the tool, in the center of the cutting edge (1200 μm from the tip) and at 200 μm from the junction between the main edge and the central edge [5,6]. These profiles are used to rebuild the 3D geometry of the cutting edge using CAD software (Figure 5). By comparison with the initial geometry of the unused drill, it is possible to evaluate the drill wear as a function of the hole number. All the new tools have an initial supposed wear *W*_0_, which is related to the cutting edge initial sharpness (edge radius of 17.5 μm).

## 3. Experimental Results

### 3.1. Axial Load in CFRP, Ti6Al4V and CFRP/Ti6Al4V Stacks

The different results obtained are illustrated in Figure 6, Figure 7, Figure 8 and Figure 9. In the CFRP plate alone, the thrust force increases with the number of holes realized (Figure 6) but the level of the load remains quite low with respect to the Ti alloy case. After 60 holes, the maximum axial load is around 250 N. In theTi alloy plate alone (Figure 7), the thrust force remains quasi-constant after 25 holes. In the stack (Figure 8), this axial load increases rapidly with the number of holes, in both titanium and CFRP. The axial load values (Figure 9) for a same contact length Lc are similar to those of the material alone for the CFRP, but increase strongly with Lc for the Ti part of the stack with respect to the Ti plate alone. The wear of the drill seems to have a stronger influence for the Ti alloy part of the stack than for the CFRP part.

### 3.2. Torque in CFRP, Ti6Al4V and CFRP/Ti6Al4V Stacks

There is only a slight evolution between the torque measured in the CFRP plate alone and the one in CFRP part of the stack (Figure 10). The evolution of the torque is linear, but the torque increase remains quite slow.

In the Ti alloy, the evolution is quite different. In the Ti alloy plate alone, the torque measured increases linearly when in the Ti alloy part of the stack—there is a rapid increase of the torque which evolves linearly after the lapping process of the first holes. After 25 holes, the torque value has increased by 70% of its initial value.

### 3.3. Temperature in CFRP, Ti6Al4V and CFRP/Ti6Al4V Stacks

The temperature measurements showed that the maximum temperature reached during the drilling is higher than 180 °C in the case of the CFRP plate alone and 400 °C in the case of the Ti-6%Al-4%V plate alone (Figure 11). But for the CFRP/Ti-6%Al-4%V stack, the maximum temperature is reached in the Ti part and is higher than 750 °C.

This means that during the drilling of the stack, in the Ti part, and depending of the cutting conditions, temperatures between 400 and 800 °C can be easily reached at the contact between the drill and the chip when in the Ti alloy plate alone—this temperature ranges from 200 to 400 °C. These results are in accordance with previous works [21,24,30].

## 4. Tool Wear in CFRP, Ti6Al4V and CFRP/Ti6Al4V Stacks

### 4.1. CFRP Alone

As mentioned by other authors [2,3,4,5,6,7,8,9], when drilling CFRP-only, the drill wear is mainly of the abrasive type (Figure 12). The cutting edge radius increases leading to edge rounding wear. The local pull-out of carbide particles or micro-chipping and micro-fractures of the drill can occur due to a local stress on the cutting edge which is too high. However, in our experiments, no major chipping was observed on the cutting edge. The cutting edge directly contacts the work material flow without the presence of a stagnation zone near the cutting edge, leading to a rapid dulling of the cutting edge [20]. Flank wear, edge and spur rounding were the major wear modes for this configuration (Figure 5 and Figure 12).

### 4.2. Ti-6%Al-4%V Alone

In this case, the cutting edge does not directly contact the work material flow, as pointed out by the presence of the stagnation zone in front of the cutting edge. The cutting edge vicinity is cleaned of titanium alloy after the different passages through the Ti-only (Figure 13 bottom left). Small amounts of adhered titanium alloy subsist all over the cutting face. There is no preference for the adhesion between alpha phase (in pink/blue) or beta phase (in orange) (Figure 13 bottom right). Flank wear was the major wear mode for this configuration (Figure 5 and Figure 13).

After 60 drilled holes, there is no evidence of cutting or chisel edges’ chipping as reported by other authors. In the drilling experiments on Ti-only of Wang et al [20], the chipping locations were mainly concentrated near the chisel edge. Moreover, various authors [31,32,33,34,35] observed severe chippings on the cutting edge near the outer cutting lips where the cutting speed is much higher (Table 2). This difference may be related to the particular geometry of the drills used in our experiments, the cutting conditions and the number of holes drilled. However, the exact mechanism of the edge chipping in the drilling of Ti-only is still unknown. When drilling Ti-only, the Ti adhesion was readily observed on the cutting edge, which causes edge chipping as the fragments of the tool material break off [31]. That means that the SD205A SECO twist drill is particularly adapted to the drilling of titanium alloys.

### 4.3. CFRP/Ti-6%Al-4%V Stacks

The cutting edge vicinity is cleaned of titanium alloy after its passage through the CFRP plate (Figure 14 and Figure 15). It subsists all over the cutting face amounts of adhered titanium alloy.

The amount of adhered titanium remains higher than in the case of drilling the Ti alloy plate alone. The passage through the stack leads quickly to edge and flank wear, edge rounding wear and the local pull-out of carbide particles or micro-chipping. Micro-fractures of the drill occur due to an overly high local stress on the cutting edge. There is no preference for the adhesion between alpha phase (in pink) or beta phase (in orange).

When the tool drills again the Ti alloy part of the stack (Figure 16), the cutting edge is loaded again with adhering titanium particles. This process occurs also on the rake and clearance faces of the drill.

### 4.4. Tool Wear Measurements

The evolution of the tool wear in each of the different configurations is given in Table 3 (the corresponding contact length on the first row and volume wear in the second row). As it can be observed in Figure 14, Figure 15 and Figure 16, for the drill which has been used in the stack the increase in wear is (i) low during the passage of the tool in the titanium of the stack, because the adherent deposits protect the tool and (ii) very important when passing into the CFRP by the brushing effect and abrasion of the fibers. The cutting edge rounding measurements after interrupted tests and acid attack—used to eliminate the titanium present on the tool—show that the wear increment in the CFRP is estimated to be near 90% and therefore 10% when passing through the titanium part of the stack, whatever the number of holes drilled in the stack.

Figure 17 gives the evolution of the increment in wear with the number of holes for the same materials thicknesses (5 mm in titanium alloy and 10 mm in CFRP) whatever the configuration (Ti-only, CFRP-only or CFRP/Ti stack). It appears that the wear in the CFRP/Ti stack is not a simple addition of the wear in the CFRP-only and the Ti-only. As the drill thermomechanical loadings are not the same in the CFRP-only, the Ti-only and the CFRP/Ti stack configurations, the wear in the stack is higher than the sum of wear in each base material.

Therefore, this result is different from the conclusion of Wang et al. [20], where a Local Wear Quantity is defined by subtracting the worn tool profiles from the new tool profiles (and corresponding to a surface area). In that case, they pointed out that the tool wear in the drilling of the CFRP/Ti stack is the sum of the edge rounding wear from drilling CFRP-only and flank wear from drilling Ti-only. This difference may be related to the use of a surface criterion in one case [20] and of a volume criterion in our case. This confirms that the interaction between the CFRP and Ti parts of the stack has a strong influence on the tool wear.

## 5. Tool Wear Analysis

### 5.1. Adhesion Path in Ti6Al4V of the Stack

The pseudo-binary CW-Ti diagram shows that the titanium has a slight solubility in tungsten carbide, about 2%at (Figure 1). The ternary W/C/Ti diagram shows that tungsten carbide is not in equilibrium with the titanium (Figure 18). The diffusion path could be CW/ (Ti,W)C/ Ti. Some studies noted temperatures between 600 and 1200 °C during the dry machining of titanium alloy [30]. The temperature measurements performed in this study confirm these data and the contact temperature between the tool and the chip is likely to range from 300 to 800 °C, depending on the cutting conditions and the material which is drilled, Ti alloy alone or CFRP/Ti stacks. Given these high temperatures, it is likely that titanium can diffuse into the tungsten carbide and form the mixed carbide (Ti,W)C.

But above 300 °C, titanium can also be in solution in the α-Co (Figure 1) or produce compounds such as Ti_2_Co, TiCo, TiCo_2_ or TiCo_3_ [13]. A chemical state analysis has been performed on a titanium chip adhered on the drill, using a Jeol JAMP-9510F Field Emission Auger microprobe, with a 10 nm probe diameter, from the carbide tool to the adhered titanium chip following a straight line L with a chemical analysis every 20 nm (Figure 19 and Figure 20).

The Auger analysis shows that titanium can react with CW to form a solid solution (Ti,W)C, or TiC carbide type (Figure 20). The presence of oxygen is also detected in lower quantity than titanium, tungsten and carbon. For temperatures above 500 °C, carbides and particularly titanium carbide become very sensitive to oxidation, allowing an oxy-carbide (Ti,C,O) to form. This may explain the erosion of the tool, because this oxy-carbide is friable.

An in-depth analysis in the cobalt phase (Figure 21 and reference of point P in Figure 19) detects the presence of titanium in very small quantities, making the formation of Ti/Co compounds very unlikely.

### 5.2. Tool Wear Mechanisms in CFRP/Ti6Al4V Stacks:

The following scenario seems to be happening:
−As it passes through the CFRP, the area around the cutting edge of the tool is cleaned by abrasion of the fibers and becomes chemically very reactive.−When passing through the titanium alloy, the temperature reaches values above 500 °C, which promotes the adhesion of titanium particles to the surface of the tool by creating (Ti,W)C carbides which oxidize quickly to give (Ti,O)C oxycarbides.−Again, in the CFRP, the abrasion of the carbon fibers removes the titanium deposits from the tool and pulls out the oxycarbides associated with the titanium deposit. The abrasion of the fibers erases the imperfections of the cutting edge giving a very reactive chemical surface again.

Most of the wear is then generated during the drilling phase trough the CFRP and because the step in the titanium has created friable intermetallics.

## 6. Tool Wear Model for the Drilling of CFRP/Ti6Al4V Stacks

The models developed are based on the phenomenological approach of Iliescu et al. [2] and take into account the drill contact length *Lc* within the different materials, the wear *W*(*i*) after each drilled hole and the axial load at the (*i* + 1)*^th^* hole which depends of the wear after the *i^th^* hole.

The general model of the wear is given by Equation (5)
(5)$$\begin{array}{c} W\left(i\right)=W_{0}+A_{CFRP}\cdot {\left[\sum_{j=1}^{i}Lc_{CFRP}\left(i\right)\cdot Fz_{CFRP}\left(i\right)\right]}^{m_{CFRP}}\\ +A_{Ti}\cdot {\left[\sum_{j=1}^{i}Lc_{Ti}\left(i\right)\cdot Fz_{Ti}\left(i\right)\right]}^{m_{Ti}} \end{array}$$
where *W*_0_ (in μm^3^) is representative of the initial 3D sharpness of the tool (defined in Figure 5), *A_sub_* and *m_sub_* are, respectively, the wear rate and the sensitivity to the wear rate in each material. As mentioned in §4.4, the overall tool wear in the CFRP/Ti stack for the *i^th^* hole is attributed for 90% at the CFRP part and for 10% at the titanium alloy part of the stack.

The axial load (in N) for the following (*i* + 1)*^th^* hole is defined by Equation (6):(6)$$F_{z}\left(i+1\right)=Kc_{mat}\cdot W{\left(i\right)}^{{}_{mat}}$$
where *Kc_mat_* is a constant parameter which depends on the drilled material, the feed rate and the cutting speed, and *n_mat_* is the sensitivity of the axial load to wear.

Table 4 gives the different values of the constant parameters, as well as the regression coefficient for the different models. Figure 22 and Figure 23 show the evolution of wear and axial load in the different materials and configurations as a function of the contact length. The models fit the experimental data very well. Since the thermomechanical loadings on the tool are different for each configuration, the coefficients *A_sub_* and *m_sub_* are different.

## 7. Discussion

This study demonstrates that improving the performance of the tool (higher tool life and productivity) involves the use of a coating, and/or the development of a tool material having low chemical affinity with Ti, and/or the use of a cryogenic lubricant.

In the first case, coatings such as AlTiN, AlCrN or diamond are effective and allow for improvement of the tool life.

In the second case, Sandvik Coromant AB has recently developed specific designed substrates that offer a very limited reaction window between the carbon from WC and titanium [36,37,38], which confirms our results. They demonstrate that the contribution of the PVD-coating to the tool performance in case of their reference GC1105 is smaller than the contribution of the substrate. They propose two variants.

The first substrate variant is a cemented carbide with a Co-depleted surface that contains “overstoichiometric” carbon on the surface [37]. The cemented carbide possesses more resistance to wear because (i) a low content of Cobalt on the surface reduces the possibility of the Titanium forming a melt with the binder phase, and (ii) extra graphite provides a source of carbon that will interact with the workpiece material (Ti-alloy) before attacking the carbon of the WC grains, retarding their dissolution and hence the wear.

The second variant (Variant 4) is a cemented carbide with a low-Co composition (3wt%) and a coarse WC particle size [38]. The cemented carbide possesses more resistance to wear because of (i) a reduced content of Cobalt on the overall composition that reduces the possibility of the Titanium to form a melt with the binder phase, and (ii) a coarse grain size that makes it more resistant to notch wear and suitable for intermittent cutting. In the third case, the cryogenic lubricant has demonstrated its effectiveness with respect to the tool wear, but it is necessary to demonstrate the type and magnitude of the generated residual stresses in the Ti part.

## 8. Conclusions

This contribution reported on a comparative carbide tool wear study based on drilling experiments on CFRP-only and Ti-only as well as CFRP/Ti stacks.
It has been confirmed the dominating tool wear modes when drilling CFRP-only (edge and spur rounding wear) or Ti-only (flank wear and edge chipping). It has been pointed out that the axial load and the temperature fields on the cutting edge are quite different and lower in the base materials than in the stack. Thus, the novelty and originality of this contribution concerns the mechanisms of wear and the diffusion path, explaining the adhesion of titanium on the tool and its posterior elimination by the carbon fibers.This diffusion is favored when drilling the stack, by the high temperature and pressure achieved on the cutting edge, leading to a strong reaction between titanium and CW particles of the tool. Due to these high temperatures reached, (Ti,W)C carbides are formed which oxidize quickly to give (Ti,O)C oxycarbides. These friable oxycarbides facilitate the CW particles shearing and pull-out (tip and edge chipping) during the CFRP drilling phase. The abrasion of the fibers erases the imperfections of the cutting edge giving, a clean and reactive chemical surface once more. This behavior explains that the dominating wear modes in the stack are a combination of chipping, edge and spur rounding.An iterative phenomenological model has been developed which takes into account all the drilling history of the tool. The model allows for the determination of the wear of the tool for the *n* holes performed and for the prediction of the axial load for the *n+1* hole, depending of the drilling conditions and the wear reached at the end of the *n^th^* hole. The wear and the axial load models developed fit the experimental data very well (error of prediction below 10%) and can be used for the monitoring of the process.The results of this contribution should make it possible to optimize the drilling of the CFRP/Titanium stack by improving the tool life. One of the solutions is to modify the chemical composition of carbides to reduce the chemical affinity between CW and titanium. Early solutions are being developed by Sandvik among others, and will have to be evaluated with tool geometries specific to this type of stacking. The second solution consists of covering these diamond-modified chemical tools and checking that the growth and adhesion of the diamond is not modified by the presence of new elements. The third solution is to limit the increase in temperature in the cutting zone by using a cryogenic refrigerant. Further tests will be necessary to verify that the state of residual stresses generated on the surface of the titanium plate are of compression and do not unduly reduce the fatigue strength of the connection.

## Figures and Tables

**Figure 1 materials-12-02843-f001:**
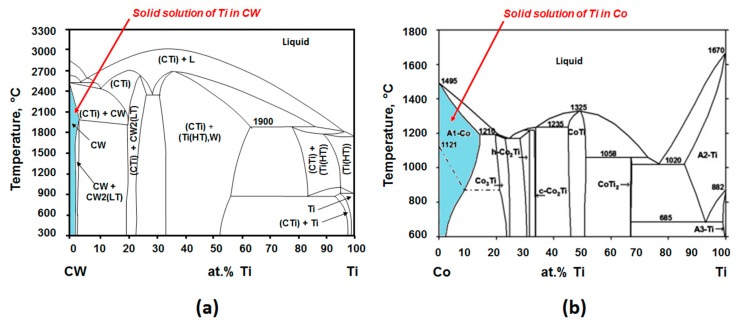
(**a**) Pseudo-binary diagram Ti/CW; (**b**) Binary diagram Ti/Co.

**Figure 2 materials-12-02843-f002:**
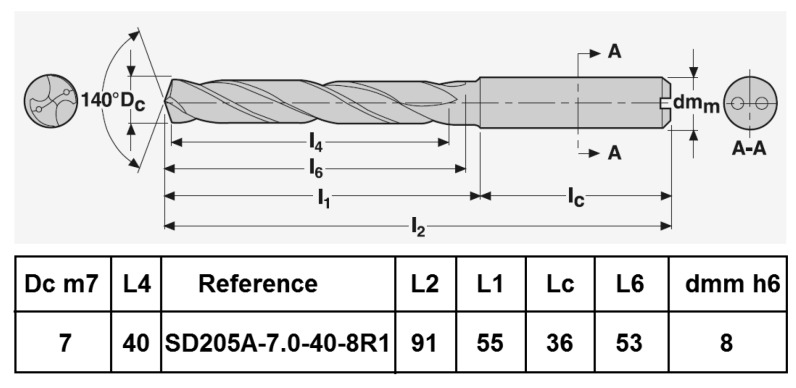
Characteristics of the drill used.

**Figure 3 materials-12-02843-f003:**
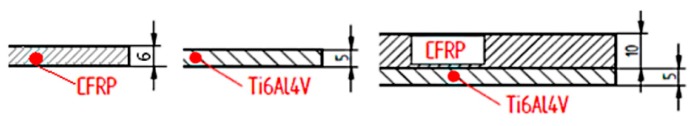
Schemes of the stacks which have been tested.

**Figure 4 materials-12-02843-f004:**
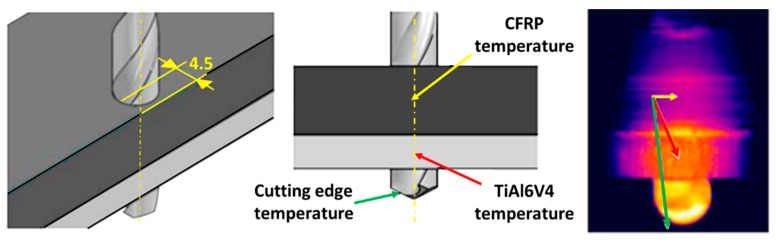
Temperature measurements in the stack and at the drill point.

**Figure 5 materials-12-02843-f005:**
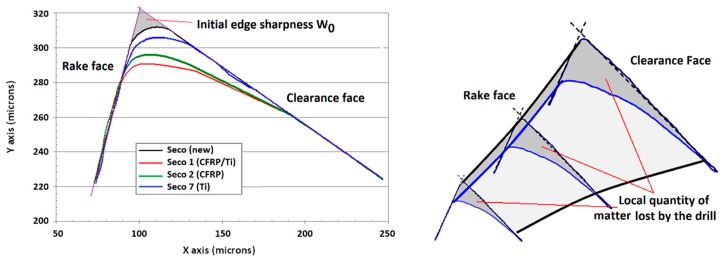
2D profile of the cutting edge measured at 1200 μm of the drill tip and reconstitution of the 3D geometry.

**Figure 6 materials-12-02843-f006:**
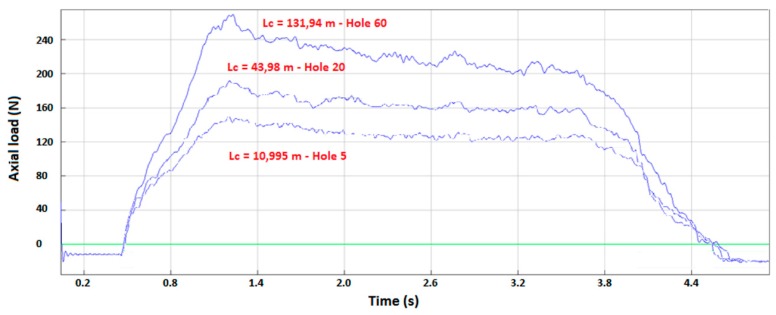
Thrust force versus time in carbon fiber composite materials (CFRP) plate alone.

**Figure 7 materials-12-02843-f007:**
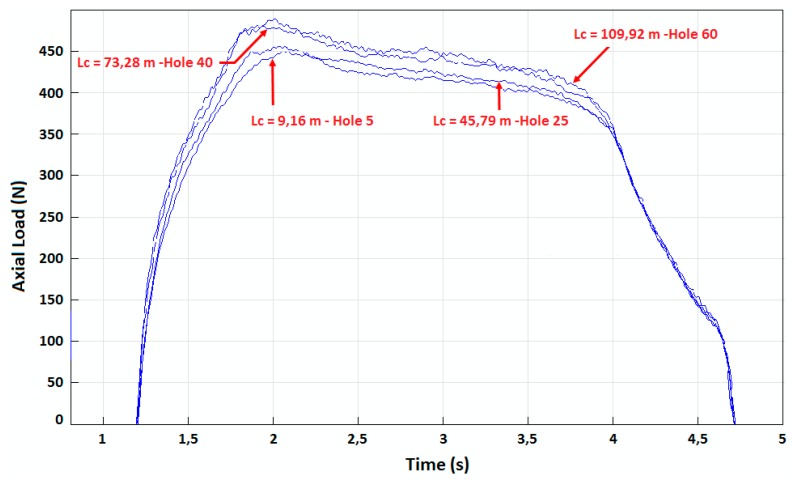
Thrust force versus time in Ti-6%Al-4%V plate alone.

**Figure 8 materials-12-02843-f008:**
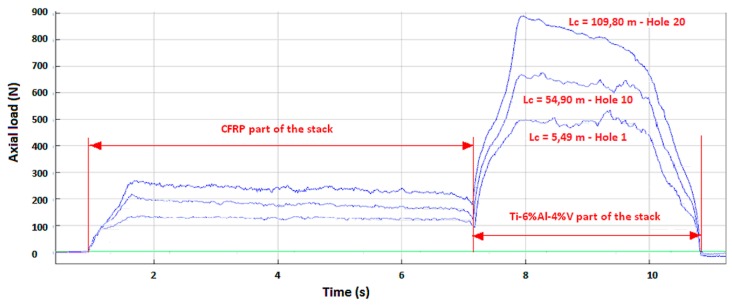
Thrust force versus time in CFRP/Ti-6%Al-4%V stacks.

**Figure 9 materials-12-02843-f009:**
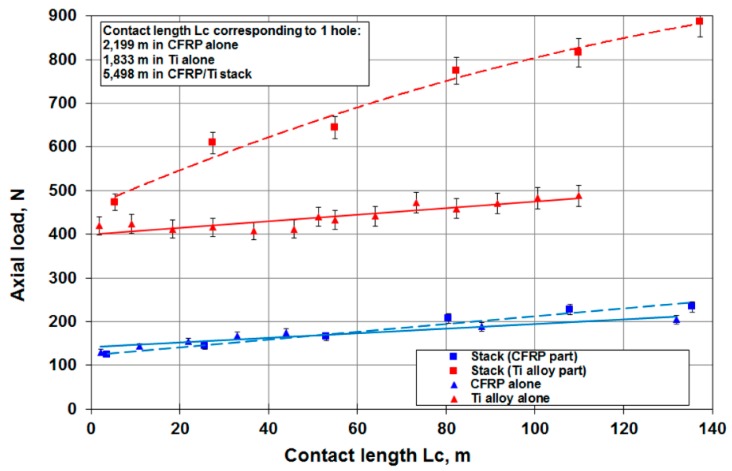
Axial load versus contact length Lc for CFRP and Ti alloy plates and the CFRP/TI alloy stack.

**Figure 10 materials-12-02843-f010:**
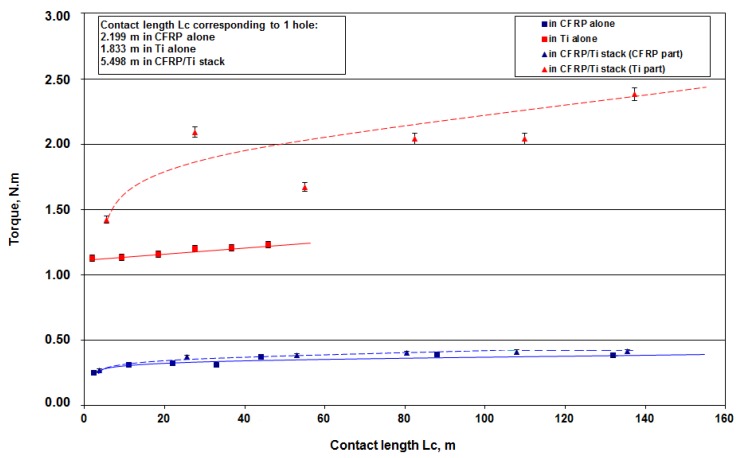
Torque versus contact length Lc in CFRP alone, in Ti-6%Al-4%V alone and in the stack (in the CFRP part and in the Ti-6%Al-4%V part).

**Figure 11 materials-12-02843-f011:**
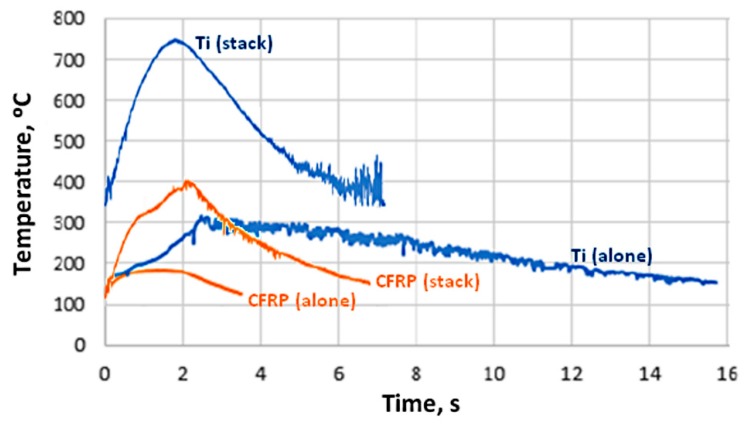
Temperature versus time in the CFRP and the Ti-6%Al-4%V plates, and in the CFRP/Ti-6%Al-4%V stacks (for hole 25).

**Figure 12 materials-12-02843-f012:**
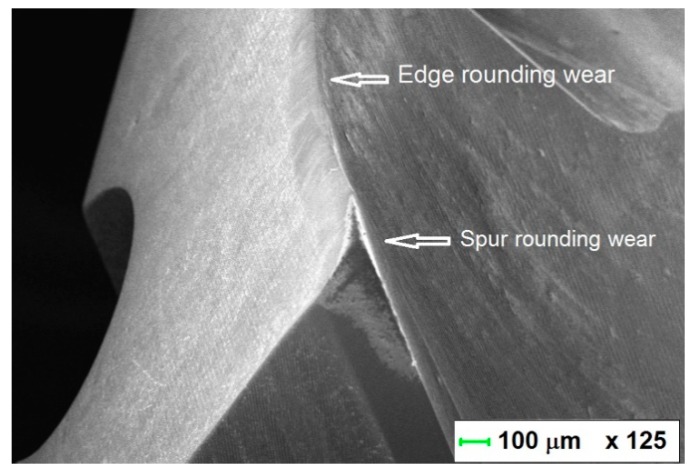
Abrasive wear of the carbide drill in the CFRP showing edge and tip rounding wear.

**Figure 13 materials-12-02843-f013:**
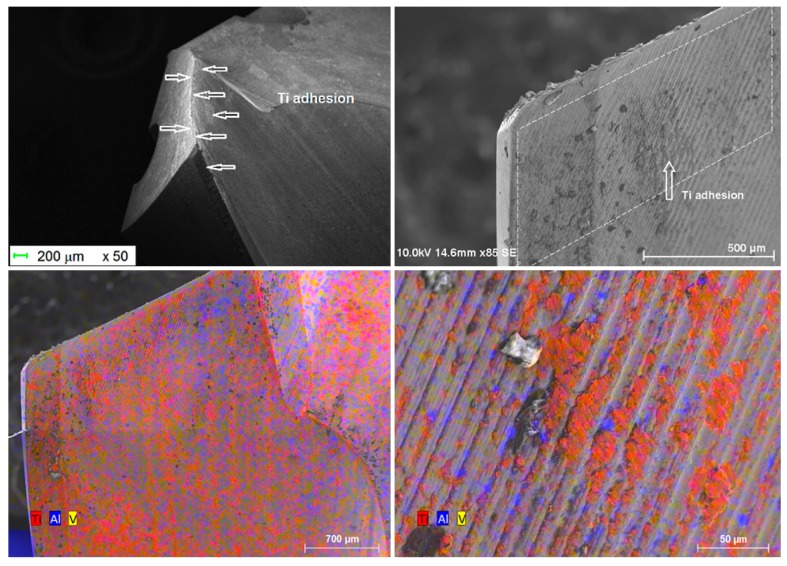
Drill aspect after 25 holes in Ti-6%Al-4%V alone.

**Figure 14 materials-12-02843-f014:**
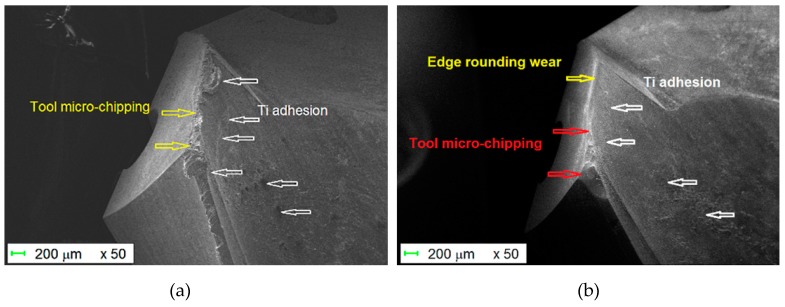
(**a**) After 25 holes in the stack, presence of adhesion of titanium alloy; (**b**) after 25 holes in the stack and passing through the CFRP, the adhered titanium has strongly decreased.

**Figure 15 materials-12-02843-f015:**
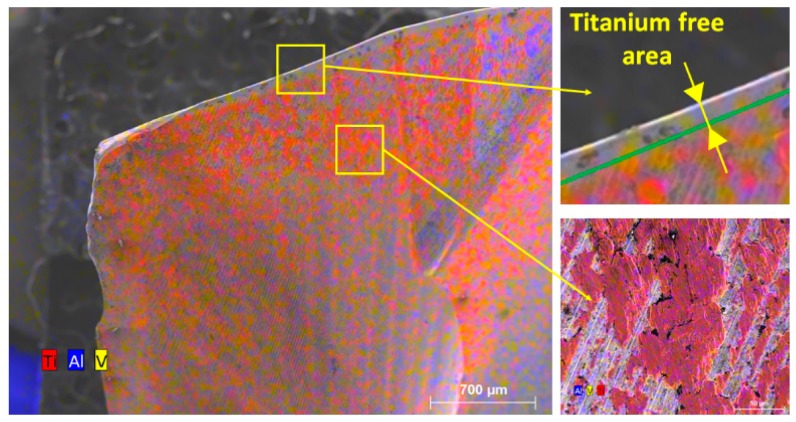
After 25 holes in the stack and passing through the CFRP, the cutting edge is cleaned from adhered titanium and some adhesion zones are still present on the cutting face. There is no evidence of predominance of α or β phases.

**Figure 16 materials-12-02843-f016:**
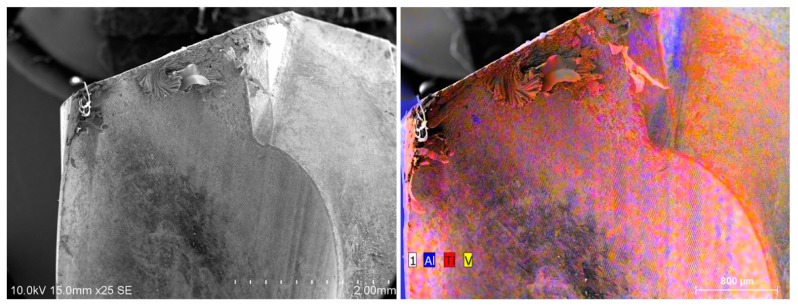
After drilling in the titanium alloy of the stack (26th hole), large adhesion zones of Ti on the tool appear again.

**Figure 17 materials-12-02843-f017:**
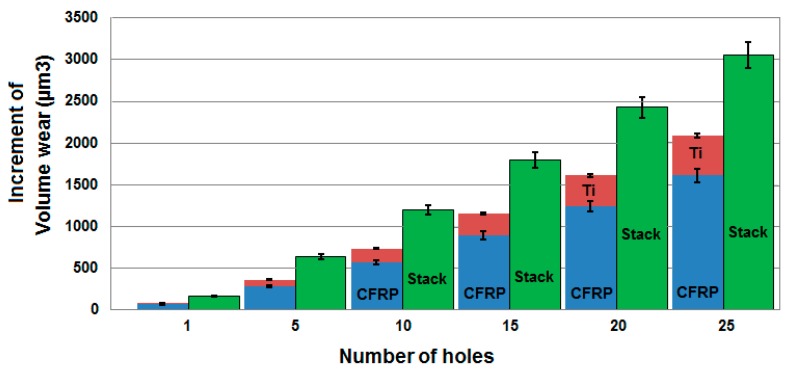
Increment of volume wear when drilling in the CFRP-only, Ti-only and CFRP/Ti stacks.

**Figure 18 materials-12-02843-f018:**
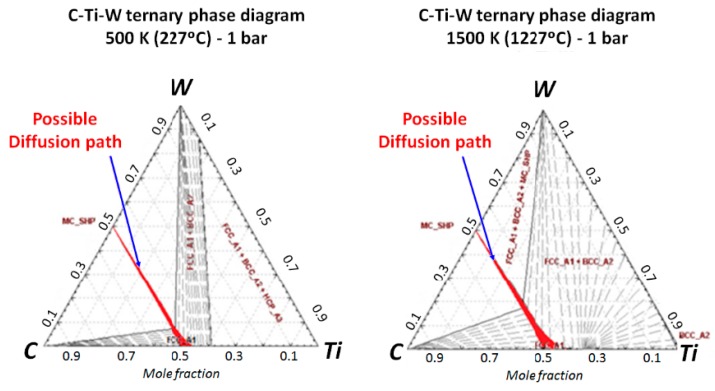
C-Ti-W ternary phase diagram at 627 °C (900 K) and 1227 °C (1500 K)

**Figure 19 materials-12-02843-f019:**
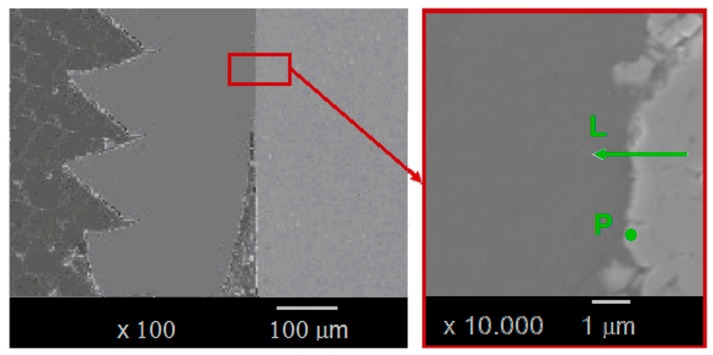
Adhered chip on the drill and zones which have been analized (composition profile along the line L and in-depth profile in point P).

**Figure 20 materials-12-02843-f020:**
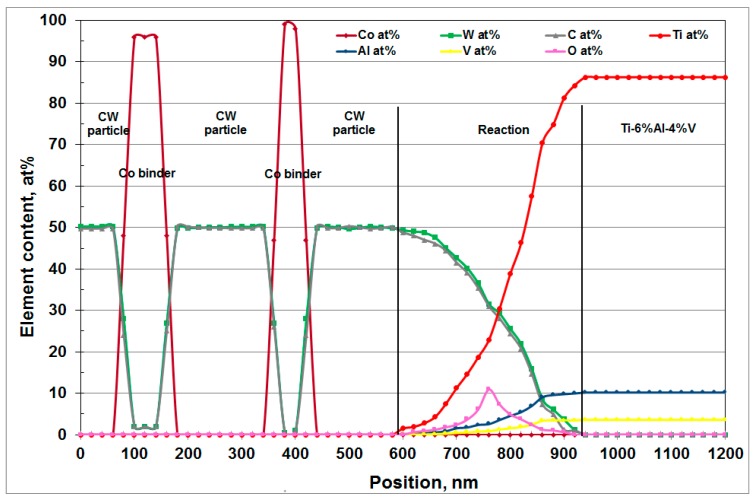
Auger microprobe analysis of the contact area between the chip and the carbide tool (line L).

**Figure 21 materials-12-02843-f021:**
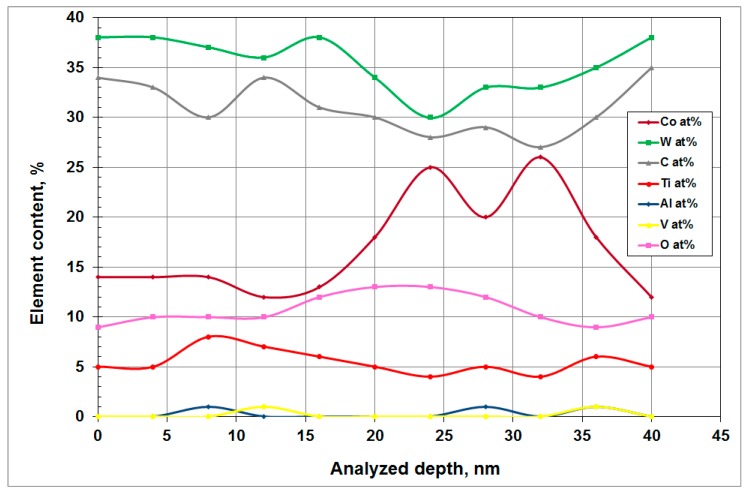
In-depth Auger analysis in the Co phase of the carbide (point P).

**Figure 22 materials-12-02843-f022:**
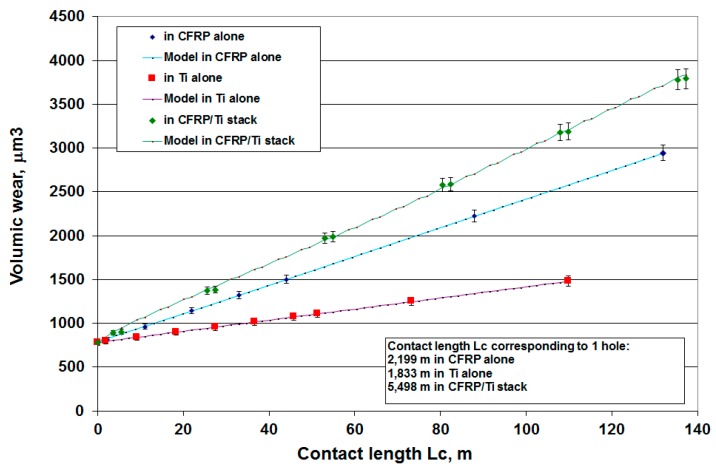
Evolution of tool wear in the different configuration and assuming for CFRP/Ti alloy stacks that 90% of the wear is produced in the CFRP and 10% in the Ti alloy.

**Figure 23 materials-12-02843-f023:**
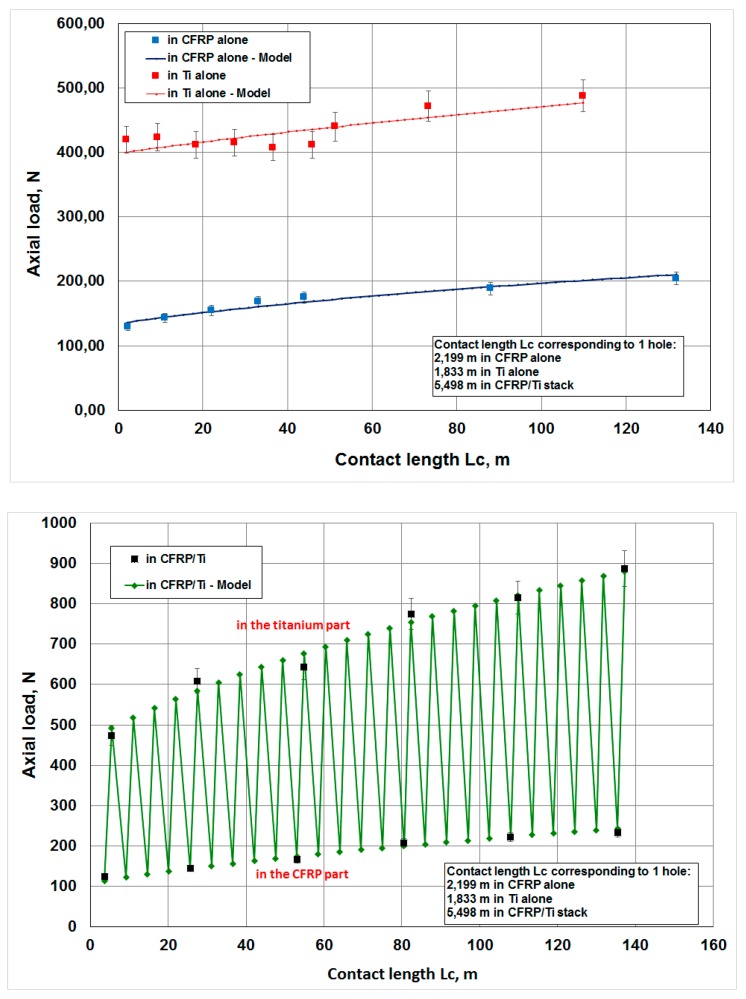
Evolution of the axial load in the different configuration and assuming for CFRP/Ti alloy stacks that 90% of the wear is produced in the CFRP and 10% in the Ti alloy.

**Table 1 materials-12-02843-t001:** Drilling parameters used in the present study.

Drill Diameter	Cutting Speed	Rotational Speed	Feed Rate	Feed Velocity	Lubricant
7 mm	40 m/min	1820 rpm	0.06 mm/rev	109 mm/min	Dry

**Table 2 materials-12-02843-t002:** Comparison of the failure modes when drilling Ti-alone plates in dry conditions and a comparison with literature references.

Reference	Vc, m/min	f, mm/rev	Tool Damage for Drilling Ti-alone in Dry Conditions with Uncoated CW-Co Drills
This work	40	0.06	Flank wear.
[20]	15	0.0508	Severe chippings at the cutting lips near the chisel edge after drilling five holes and catastrophic failure of the drill happened after drilling seven holes.
[31]	25 to 55	0.06	Rapid wear at all cutting speeds tested with similar modes of damage: non-uniform flank wear, chipping and failure.
[32]	18.3 183	0.051 0.051	N/A Drill breakage after 10 holes due to chip welding.
[33]	50	0.07	Progressive loss of TiN coating of drill and work-piece material adhesion in rake surface. Diffusion of Ti alloy in rake surface and drill helical flute were observed. Attrition in the helical flute combined with diffusion of alloy in the tool resulted in the nucleation and growth of craters.
[34]	Up to 100	Up to 0.25	Flank wear was the dominant failure mode but with TiN coated tools in milling.
[35]	55 to 100	0.1 and 0.15	Flank wear pattern under all conditions but with TiN coated tools in milling.

**Table 3 materials-12-02843-t003:** Evolutions of the drill wear in the different materials (for one cutting edge).

Lc (m) *Wear (μm^3^)*	New	5 Holes	10 Holes	15 Holes	20 Holes	25 Holes	60 Holes
**Tool SECO1**	0	27.490	54.980	82.470	109.960	137.450	
**(CFRP/Ti stack)**	783	1386	1987	2589	3191	3792	
**Tool Seco2**	0	10.995	21.990	32.985	43.980	54.975	131.940
**(CFRP alone)**	783	962	1144	1324	1504	1684	2944
**Tool SECO7**	0	9.165	18.330	27.495	36.660	45.825	109.980
**(Ti alone)**	783	842	901	959	1017	1076	1485

**Table 4 materials-12-02843-t004:** Coefficients of the models of wear and thrust force (for SECO1, upper values of Kc_mat_ and n_mat_ correspond to the passage in the CFRP when the lower values are related to the Ti alloy; r is the regression coefficient).

Tool Reference Heading	W_0_	A_CFRP_	m_CFRP_	A_Ti_	m_Ti_	*r*	Kc_mat_	n_mat_	*r*
**Tool SECO1** **(CFRP/Ti stack)**	783.84	0.7834	0.8249	0.2315	0.7672	0.999	4.1613 31.4144	0.4954 0.4047	0.989 0.998
**Tool Seco2** **(CFRP alone)**	783.84	0.2758	0.8916	0	0	0.999	14.915	0.3319	0.974
**Tool SECO7** **(Ti alone)**	782.13	0	0	0.019	0.976	0.999	63.138	0.2773	0.963

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
