# Peer review of "Wear Mechanisms and Wear Model of Carbide Tools during Dry Drilling of CFRP/TiAl6V4 Stacks"

_materials, 2019, doi:10.3390/ma12182843_

Round 1
Reviewer 1 Report
Prediction of tool life in machining process is still an issue, especially for stacks of different materials. One of such a sandwich material is carbon fiber composite material (CFRP) on TiAl6V4, being used in aeronautical assemblies. The layer of titanium alloy previewed under CFRP piece conducts to better quality of machined holes by drilling, eliminating the bad finish of the holes and the quick wear of tools.
The paper determined the wear modes of tool, the axial force, and temperature variation in the afore mentioned situation (drilling of stack materials). A semi-empirical variation law of tool wear is proposed as a function of contact length, axial force at each machining step, and some parameters established for every material (depending also on tool wear).
I consider the paper original and interesting. Also, I appreciate the huge carried out experimental work. The Introduction section is a little too extended. However, I consider that this is welcome for the sake of better understanding.
I have some suggestions and questions addressed to the authors, grouped in 8 points of major revision.
Major revision:
1. The passing of tool through titanium alloy alone, requested increased axial force and torque in comparison with CFRP. Nevertheless, from wear viewpoint this is presented as a positive aspect. The authors supposed that the drilling of titanium alloy from the stack produces only 10% of the tool wear. Is this ratio supported by their experimental results, or just an approximation? The results presented in Figure 17 seem to indicate more than 10% of wear in titanium.
2. The adhesion of titanium alloy particles protect the tool, but this fact must be correlated also with the quality of the machined surfaces. Did the authors supervise the quality of the drilled holes? Was this affected by increased tool wear?
3. The data are missing for 60 holes drilling of stack (tool SECO1). More details on Table 2 should be offered to the reader. What first and 2nd rows values mean? The value of 783 for a new tool is W0 (the sharpness of the tool)? How was it determined?
4. The same question arises for the values of different parameters of wear model, mentioned in Table 3. Were these values chosen only to fit the experimental results, or they were experimentally approximated? If these values are taken from other similar references, the paper should mention it.
5. It was asserted that by "passing of tool through the CFRP the area around the cutting edge of the tool is cleaned by abrasion of the fibers and becomes chemically very reactive" (lines 349, 355, and 399). This issue should be argued more. It is evident that abrasion mechanically sharpen and clean the tool, but it is not obvious how the tool cutting edge is chemically activated.
6. To argue the results presented below Figure 10, which are different from those of other authors, provide a supplementary table containing drilling parameters used by those researchers [28-32] which observed chisel edges chipping in cutting of Ti alloy alone.
7. The concluding phrases from lines 406 to 426 have to be included in Introduction section, as these assumptions and comments do not reflect the findings of the paper, being rather a bibliographical study.
8. Line 371: Figures 22 and 23 show the evolution of wear and axial load, not only of axial load.
Author Response
The passing of tool through titanium alloy alone, requested increased axial force and torque in comparison with CFRP. Nevertheless, from wear viewpoint this is presented as a positive aspect. The authors supposed that the drilling of titanium alloy from the stack produces only 10% of the tool wear. Is this ratio supported by their experimental results, or just an approximation? The results presented in Figure 17 seem to indicate more than 10% of wear in titanium.
This ratio has been determined experimentally. For some drills, after 10 and 25 holes, a cleaning has been performed in a circulating stirred bath of HNO3 and HF acids (with HNO3 to HF ratio of 2 to 4), at a temperature of 40ºC in order to dissolve the titanium present on the surface of the tool. The profile of the cutting edge was measured as described in section 2.4 and determined wear. The tool then made a hole in the CFRP and after cleaning the profile was measured and the wear defined. It was thus determined that proportion of 10% and 90% respectively during the transition in titanium and CFRP. This is because although damaged by the diffusion of titanium into the CW and formation of friable oxycarbides, it is during the passage in the CFRP that these oxycarbides are removed from the surface of the tool. This proportion is an estimate because based on 2 measurements (after 10 and 25 holes).
But this proportion has also been confirmed during the fitting of the experimental results in order to determine the values of the different coefficients of the model. A proportion 10/90 results in a minimization of the error of prediction.
The adhesion of titanium alloy particles protect the tool, but this fact must be correlated also with the quality of the machined surfaces. Did the authors supervise the quality of the drilled holes? Was this affected by increased tool wear?
Yes, roughness, burr height and diameter accuracy of the holes have been measured. But it was not the objective of the study which was to understand the wear mechanisms. For this reason, we have not added information about hole quality.
In titanium plates, the burr height remains constant up to 60 holes (175 microns). In the stack, the burr height depends on the tool wear and increases from 200 microns (1 hole) to 600 microns for the hole 25.
In titanium or CFRP plates, the roughness Ra remains constant at respectively 0.6 and 0,5 microns for the 60 holes. In the stack, the roughness is again highly dependent of the wear (2.5 microns for the first hole and 8 microns for hole 25).
Regarding the diameter accuracy, there is no difference between the upper and lower zone of the hole for the titanium plates and for the 60th first holes this diameter is quite constant at 7,015±0,005 mm. For the CFRP plates, the diameter is constant along the hole at the beginning (6,980±0,005 mm) but diverges with the number of drilled holes (6,995±0,005 mm at the top and 7,005±0,005 mm at the bottom). In the case of the stack, the diameter remains constant in the titanium (7,015±0,005 mm) while there is a difference in the CFRP between the top and the bottom since the first hole: respectively 6,990±0,005 mm and 7, 205±0,005 mm for the first hole; 7,025±0,005 mm and 7,500±0,005 mm for the 25th hole.
The data are missing for 60 holes drilling of stack (tool SECO1). More details on Table 2 should be offered to the reader. What first and 2nd rows values mean? The value of 783 for a new tool is W0 (the sharpness of the tool)? How was it determined?
Tests have been performed to reach a similar contact length for the 3 materials. As in the study we have also performed vibration assisted drilling and cryogenic drilling, it was decided that 25 holes in the stack was sufficient for these trials. Do not forget that the drilling conditions are also extreme (cutting speed of 40 m/min in dry condition instead of an usual 20 m/min in lubricated condition).
Table 2 gives the volume wear (2nd row) for one cutting edge and the corresponding contact length (1st row).
The value of 783 μm3 corresponds to the initial sharpness of the tool (W0). It is the volume between the profile of the new cutting edge and the planes containing the rake face and the clearance face of the tool (figure 5 left for a 2D view). This value has been determined for all the drills for each of the 3 configurations (10 drills per configuration) and the mean value calculated leading to 783±35 μm3 (for the 30 drills). We can have a slight variation of this value for each of the configuration as occurred in Table 3.
The same question arises for the values of different parameters of wear model, mentioned in Table 3. Were these values chosen only to fit the experimental results, or they were experimentally approximated? If these values are taken from other similar references, the paper should mention it.
In Table 3, the value of W0 has been determined as previously explain. The other values have been determined by multilinear regression, minimizing the error of prediction. In that case, we have added the coefficient of regression of the model.
It was asserted that by "passing of tool through the CFRP the area around the cutting edge of the tool is cleaned by abrasion of the fibers and becomes chemically very reactive" (lines 349, 355, and 399). This issue should be argued more. It is evident that abrasion mechanically sharpen and clean the tool, but it is not obvious how the tool cutting edge is chemically activated.
Fig 15 points out a zone free of titanium near the cutting edge where the tool wear is located. That means that this area is similar to the material of an unused drill and that its chemical reactivity is higher than other parts of the rake face for example which are protected by the remaining titanium.
To argue the results presented below Figure 10, which are different from those of other authors, provide a supplementary table containing drilling parameters used by those researchers [28-32] that observed chisel edges chipping in cutting of Ti alloy alone.
The following table has been added.
Reference |
Vc, m/min |
f, mm/rev |
Tool damage for drilling Ti-alone in dry conditions with uncoated WC-Co drills |
This work |
40 |
0.06 |
Flank wear. |
[20] |
15 |
0.0508 |
Severe chippings at the cutting lips near the chisel edge after drilling five holes and catastrophic failure of the drill happened after drilling seven holes. |
[28] |
25 to 55 |
0.06 |
Rapid wear at all cutting speeds tested with similar modes of damage: non-uniform flank wear, chipping and failure. |
[29] |
18.3 183 |
0.051 0.051 |
N/A Drill breakage after 10 holes due to chip welding. |
[30] |
50 |
0.07 |
Progressive loss of TiN coating of drill and work-piece material adhesion in rake surface. Diffusion of Ti alloy in rake surface and drill helical flute were observed. Attrition in the helical flute combined with diffusion of alloy in the tool resulted in the nucleation and growth of craters. |
[31] |
Up to 100 |
Up to 0.25 |
Flank wear was the dominant failure mode but with TiN coated tools in milling. |
[32] |
55 to 100 |
0.1 and 0.15 |
Flank wear pattern under all conditions but with TiN coated tools in milling. |
The concluding phrases from lines 406 to 426 have to be included in Introduction section, as these assumptions and comments do not reflect the findings of the paper, being rather a bibliographical study.
This information has been added in a discussion section in order to confirm the conclusions of our study. These carbides are not currently marketed but can be an interesting solution for the future. We are in contact with the supplier to test this material in the stacks.
Line 371: Figures 22 and 23 show the evolution of wear and axial load, not only of axial load.
Correction made.
Reviewer 2 Report
The present work deals with the wear of carbide tools for CFRP and titanium drilling. In particular, the authors carried out an experimental campaign, in which they drilled CFRP and titanium both alone and stacked, measuring the temperature, the forces and the torque and evaluating the tool wear. The topic is interesting and the performed experimental analysis appears quite exhaustive. Due to this reason, the referee is positive toward this submission; nevertheless, in referee’s opinion there is still room for some improvements, and the following minor revisions are warmly encouraged:
Page 3 line 122: is there a reference between square brackets? Please check. How were the CFRP and the titanium tied together to form the stack? Were they bonded or co-cured? Please give a short description of the manufacturing process. “Author contributions” and “funding” seem to be incorrect. Please complete these paragraphs in a proper way. Some interesting works are missing among the references. The authors are invited to read the following papers:
“Zitoune, R., Krishnaraj, V., Collombet, F., Le Roux, S. Experimental and numerical analysis on drilling of carbon fibre reinforced plastic and aluminium stacks (2016) Composite Structures, 146, pp. 148-158. DOI: 10.1016/j.compstruct.2016.02.084” and
“Sorrentino, L., Turchetta, S., Colella, L., Bellini, C. Analysis of Thermal Damage in FRP Drilling (2016) Procedia Engineering, 167, pp. 206-215. DOI: 10.1016/j.proeng.2016.11.689”.
Author Response
Page 3 line 122: is there a reference between square brackets? Please check.
Correction made.
How were the CFRP and the titanium tied together to form the stack? Were they bonded or co-cured? Please give a short description of the manufacturing process.
The CFRP and titanium plates are neither bonded nor co-cured. Details are given in the corrected paper.
“Author contributions” and “funding” seem to be incorrect. Please complete these paragraphs in a proper way.
Corrections made.
Some interesting works are missing among the references. The authors are invited to read the following papers: “Zitoune, R., Krishnaraj, V., Collombet, F., Le Roux, S. Experimental and numerical analysis on drilling of carbon fibre reinforced plastic and aluminium stacks (2016) Composite Structures, 146, pp. 148-158. DOI: 10.1016/j.compstruct.2016.02.084” “Sorrentino, L., Turchetta, S., Colella, L., Bellini, C. Analysis of Thermal Damage in FRP Drilling (2016) Procedia Engineering, 167, pp. 206-215. DOI: 10.1016/j.proeng.2016.11.689”.
These papers have been added to the bibliography and 2 paragraphs added in the introduction to explain these research works.
Reviewer 3 Report
I recommend the title of the article as follows: Wear mechanisms and wear model of carbide tools during dry drilling of composite (CFRP / TiAl6V4) plates.
In the annotation, it is necessary to clearly state the novelty of the research, add information about the tool wear model and the accuracy of its prediction, as well as briefly state the practical significance of the work.
Figure 1 needs to be eliminated from a typo: it is necessary to interchange the pictures and insert the notation “a” and “b”. In order to improve the perception of the material in Figure 4, it is necessary to sharpen the pictures, and Figure 5 and 18 should be enlarged.
Conclusions rewrite more concisely. In the first conclusion, determine the novelty, i.e. what was first received. In the second and third conclusions, we list in detail new data on the mechanism and model of wear of a carbide tool (it is necessary to indicate the accuracy of model prediction). In the fourth conclusion, note the practical significance of the work and briefly the prospects for future research related to improving the efficiency of drilling composite (CFRP / TiAl6V4) plates.
After correcting all comments, the article may be accepted in print.
Author Response
I recommend the title of the article as follows: Wear mechanisms and wear model of carbide tools during dry drilling of composite (CFRP / TiAl6V4) plates.
After discussion among the authors, it seems to us that the initial title is more representative of our work by referring to composite / metal stacks.
In the annotation, it is necessary to clearly state the novelty of the research, add information about the tool wear model and the accuracy of its prediction, as well as briefly state the practical significance of the work.
Modifications and additions made in the text.
Figure 1 needs to be eliminated from a typo: it is necessary to interchange the pictures and insert the notation “a” and “b”.
Correction made.
In order to improve the perception of the material in Figure 4, it is necessary to sharpen the pictures, and Figure 5 and 18 should be enlarged.
Modifications made.
Conclusions rewrite more concisely. In the first conclusion, determine the novelty, i.e. what was first received. In the second and third conclusions, we list in detail new data on the mechanism and model of wear of a carbide tool (it is necessary to indicate the accuracy of model prediction). In the fourth conclusion, note the practical significance of the work and briefly the prospects for future research related to improving the efficiency of drilling composite (CFRP / TiAl6V4) plates.
The conclusions have been rewritten following the instructions of the reviewer.
Round 2
Reviewer 1 Report
The authors responded to all the questions and made the requested amendments.
I warmly recommend this paper for publication in Materials journal and I congratulate the authors for the good work.